# OpenReview forum: "Toward Stable Value Alignment: Introducing Independent Modules for Consistent Value Guidance"
_ICML.cc/2026/Conference — ICML 2026 spotlight_

### Official Review · Reviewer_JHU4 · 2026-03-11

**Soundness:** 2
**Presentation:** 3
**Significance:** 3
**Originality:** 3
**Overall Recommendation:** 4
**Confidence:** 5

**Summary:**

This paper proposes SVGT (Stable Value-Guided Tuning), a method that aims to improve the controllability and alignment stability of large language models by constructing a Stable Value Space at intermediate layers and applying value-guided corrections during generation. The core idea is to derive a value representation subspace from intermediate hidden states and use it to guide the model’s generation trajectory. The authors claim that this design improves alignment stability while preserving the model’s task capabilities. Experiments are conducted on several alignment-related benchmarks to demonstrate improvements in safety and robustness.

**Compliance With Llm Reviewing Policy:**

Affirmed.

**Final Justification:**

The rebuttal directly addresses my main concerns. These clarifications largely resolve my doubts about the core methodological claims and empirical support, so I raised my score.

**Key Questions For Authors:**

1. How is the stability of the proposed Stable Value Space defined and quantitatively measured?
2. Does the proposed intervention affect the model’s general capabilities? Please report results on standard benchmarks such as MMLU, GSM8K, and HumanEval.
3. If $H^(l*)$ includes both prompt and generated tokens, how does the aggregation $A(H^(l*))$ avoid entangling prompt semantics with the value representation?
4. How sensitive is the method to the choice of aggregation operator and intermediate layer?

**Limitations:**

Yes.

**Strengths And Weaknesses:**

Strengths
1. The paper studies an important problem in LLM alignment, namely the instability of value representations during generation. Exploring intermediate-layer value representations for guiding generation is an interesting direction that attempts to decouple alignment signals from the model’s task representations.
2. The proposed method is relatively modular and lightweight. Since the value space is constructed from intermediate hidden states, the framework could potentially be applied to various decoder-only architectures without major architectural modifications.
3. The paper attempts to introduce a more interpretable alignment mechanism by explicitly defining a value representation space and using it to guide generation, which could provide insights into internal alignment dynamics.

Weaknesses

1. The concept of “Stable Value Space” is insufficiently validated, which is my main concern. The paper positions the Stable Value Space as the central concept of the method, but it does not provide quantitative evidence demonstrating why or in what sense this space is stable. For example, the authors do not analyze the variance of value representations across prompts, training steps, or layers. As a result, the stability claim remains largely conceptual.
2. Limited evaluation on standard capability benchmarks. The experiments mainly focus on alignment or safety metrics. However, the paper does not report results on widely used capability benchmarks such as MMLU, GSM8K, or HumanEval. Without such evaluations, it is difficult to determine whether the proposed intervention negatively affects the model’s general reasoning and knowledge capabilities.
3. Potential entanglement between prompt semantics and value representation. In the value space construction, the hidden states are defined as , and the value state vector is computed as , while the prompt context vector is defined as . If includes both prompt tokens and generated tokens, the aggregation used to construct will inevitably incorporate prompt information. This raises a conceptual concern that the resulting representation may largely reflect the overall sequence representation rather than an independent value-related subspace.
4. Sensitivity of the aggregation operator is not analyzed. The value vector is obtained through an aggregation operator , such as last-token selection or attention pooling. However, the paper does not analyze how sensitive the method is to the choice of aggregation strategy. Different aggregation operators may lead to significantly different value representations.
4. Layer selection lacks justification. The value space is constructed from a specific intermediate layer , but the paper does not provide sufficient justification for choosing this layer. Since representations vary significantly across layers in large language models, the effectiveness of the method may depend heavily on this design choice.

---

> ### Author Rebuttal · Authors · 2026-03-30
>
> We sincerely appreciate your insightful comments on our work! Below are our responses to the issues you have raised. All experiments below use Llama-3.2-3B as the primary backbone unless otherwise noted.
>
> ---
>
> **Q1: How is the stability of the Stable Value Space quantitatively measured? (W1)**
>
> Thanks for this important question. We define **stability** as the consistent ability to capture value signals within dynamic residual streams across diverse contexts. Concretely, a stable value space exhibits **structural consistency**: despite drastic contextual variations, the core geometric structure (the decision boundary separating “harmful” from “harmless”) remains largely invariant, ensuring reliable guidance signals. We report (i) OOD separability of the value representation and (ii) OOD behavioral mitigation:
>
>  (i) **Recognition stability.** Table A compares our value encoder to a linear probe trained with the same data/config. On unseen ToxicChat[1], the linear probe degrades substantially , while the value encoder retains higher separability, showing that its value decision boundary is structurally stable across unseen contexts.
>
>
> *Table A: Value encoder vs. linear probe (1000 samples per evaluation set)*
> | Dataset | Method | Acc | Macro F1 | AUROC |
> | --- | --- | --- | --- | --- |
> | In-Dist. | Value Encoder | 0.896 | 0.889 | 0.963 |
> | In-Dist. | Linear Probe | 0.808 | 0.793 | 0.880 |
> | ToxicChat (OOD) | Value Encoder | 0.895 | 0.770 | 0.794 |
> | ToxicChat (OOD) | Linear Probe | 0.763 | 0.508 | 0.554 |
>
> (ii) **Intervention stability.**  Table B shows that on unseen XSTest [2] and ToxicChat [1] prompts, SVGT consistently reduces ASR and increases refusal rate, indicating that guidance remains effective under distribution shift.
>
> *Table B: ASR & Refusal Rate on OOD adversarial datasets (100 samples per dataset)*
> | Dataset | Method | ASR | Refusal Rate |
> | --- | --- | --- | --- |
> | XSTest | No Guidance | 42% | 35% |
> | XSTest | SVGT | 22% | 61% |
> | ToxicChat | No Guidance | 67% | 17% |
> | ToxicChat | SVGT | 23% | 66% |
>
> ---
>
> **Q2: Does the proposed intervention affect the model’s general capabilities? (W2)**
>
> We report MMLU[3] (5-shot, 1000-question subset) and GSM8K[4] (8-shot CoT, 500-question subset): SVGT causes small drops (55%→53%; 64%→61%; Table C). HumanEval is omitted due to time constraints.
>
> *Table C: Utility benchmarks*
> | Benchmark | Baseline | SVGT |
> | --- | --- | --- |
> | MMLU (5-shot, 1000-Q subset) | 55% | 53% |
> | GSM8K (8-shot CoT, 500-Q subset) | 64% | 61% |
>
> ---
>
> **Q3: If $H^{(l)}$ includes both prompt and generated tokens, how does the aggregation $A(·)$ avoid entangling prompt semantics with the value representation? (W3)**
>
> We note that $h_v$​ inevitably contains prompt information, which is precisely why we designed the dual-pathway architecture (Sec 3.1): While the unconditional pathway captures explicit value cues in the text (e.g., surface harmful patterns), the conditional pathway explicitly treats the prompt as a condition and uses cross-attention to judge if the response reflects genuinely harmful compliance (Eq. 2). This separation lets SVGT distinguish ‘mentioning a risky topic’ from ‘complying with a risky request’ rather than conflating prompt with value.
>
> ---
>
> **Q4: How sensitive is the method to the choice of aggregation operator and intermediate layer? (W4 + W5)**
>
> **Sensitivity of aggregation operator.** We conducted an analysis between last-token and attention pooling across model scales on WildGuardMix (Stage 1 training on a 50000-sample subset,  1 epoch):
>
> *Table D: Aggregation operator ablation*
> | Backbone | Aggregation | AUROC |
> | --- | --- | --- |
> | GPT-2  | Last-token | 72.30 |
> | GPT-2  | Attention pooling | 73.91  |
> | Llama-3.2-3B | Last-token | 83.69 |
> | Llama-3.2-3B | Attention pooling | 87.00 |
>
> GPT-2 is nearly agnostic; Llama-3.2-3B gains +3.31 AUROC points with pooling. The rationale is that smaller, shallower models concentrate contextual information in the final token position, while larger models distribute information more broadly, requiring attention pooling for comprehensive capture.
>
> **Layer selection.** Figure 9 (Appendix D.1) provides systematic justification. We evaluated all layer positions on GPT-2 and found that middle-to-late layers achieve optimal discrimination performance, consistent with prior findings [5] that value-relevant semantic structures emerge in deeper layers.
>
> ---
>
> **References**
>
> [1] Z. Lin et al. *ToxicChat: Unveiling Hidden Challenges of Toxicity Detection in Real-World User-AI Conversation*. EMNLP, 2023.
>
> [2] P. Röttger et al. *XSTest: A Test Suite for Identifying Exaggerated Safety Behaviours in Large Language Models*. NAACL, 2024.
>
> [3] D. Hendrycks et al. *Measuring Massive Multitask Language Understanding*. ICLR, 2021.
>
> [4] K. Cobbe et al. *Training Verifiers to Solve Math Word Problems*. arXiv:2110.14168, 2021.
>
> [5] K. Park et al. *The Linear Representation Hypothesis and the Geometry of Large Language Models*. ICML, 2024.

---

> > ### Author Rebuttal · Reviewer_JHU4 · 2026-04-01
> >
> > The authors have addressed my main concerns.

---

> > > ### Author Response · Authors · 2026-04-03
> > >
> > > We sincerely thank you for confirming that our responses have addressed your concerns. Your questions have driven substantive improvements that will be reflected in the revision. We are grateful for your recognition and look forward to incorporating your suggestions into the final version.

---

### Official Review · Reviewer_eU5u · 2026-03-12

**Soundness:** 3
**Presentation:** 3
**Significance:** 3
**Originality:** 4
**Overall Recommendation:** 4
**Confidence:** 3

**Summary:**

This paper proposes the Stable Value Guidance Transformer (SVGT), an inference-time alignment framework that decouples value modeling from the LLM backbone. The method builds a separate value space using dedicated encoders, computes a correction signal in that space with a learned discriminator, and injects the resulting guidance into generation through dynamically updated Bridge Tokens rather than directly modifying backbone activations. The paper evaluates SVGT on several backbones and safety benchmarks, including HarmBench, and reports substantial reductions in harmfulness and attack success rate while largely preserving language modeling performance.

**Compliance With Llm Reviewing Policy:**

Affirmed.

**Key Questions For Authors:**

1. Can the authors provide stronger externalized evaluation beyond the learned discriminator, for example using an independent safety classifier, LLM-as-judge, or human audit? This would help address concerns about circularity.

2. Can the authors report broader utility-preservation metrics, such as instruction-following, reasoning, or safe-helpful tradeoff evaluations, to clarify whether the method reduces harmfulness without causing over-refusal?

3. Why are important inference-time baselines such as PPLM, GeDi, DExperts, or robust decoding approaches not included? A comparison against these methods would significantly strengthen the paper’s empirical positioning.

**Limitations:**

yes

**Strengths And Weaknesses:**

Strengths

1. The paper proposes a genuinely interesting architectural idea. The core design choice—keeping value modeling in an independent module and coupling it to the backbone through Bridge Tokens rather than direct residual editing—is novel and well motivated. This gives the paper a clearer architectural identity than many prior inference-time alignment methods.

2. The method is thoughtfully designed and reasonably well supported by ablations. In particular, the comparison between Bridge Token guidance and direct injection helps justify the main mechanism, and the EMA-based refresh design is a sensible way to balance stability and responsiveness during decoding. The overall system design feels coherent rather than ad hoc.

3. The empirical results are promising. The paper evaluates multiple backbones and includes adversarial testing on HarmBench, which is more meaningful than relying only on internal safety scores. The reported reductions in ASR are substantial, and the efficiency measurements are also useful for understanding the practical trade-offs of the method.

Weaknesses

1. The evaluation still depends too heavily on the paper’s own learned discriminator. Since the discriminator is central both to training and to several reported safety outcomes, there is a real concern about circularity or reward hacking. I would be much more convinced with stronger externalized evaluation, such as independent classifiers, broader LLM-as-judge evaluation, or small-scale human auditing.

2. Capability preservation is not evaluated broadly enough. The paper mostly uses perplexity to argue that utility is preserved, but this leaves open the possibility of over-refusal or degraded helpfulness on benign prompts. I would have liked to see a more diverse set of utility metrics, such as instruction-following, reasoning, or safe-helpful tradeoff evaluations.

3. The baseline coverage is incomplete for a paper in this area. In particular, several highly relevant inference-time baselines such as PPLM, GeDi, DExperts, or more robust decoding methods are not included experimentally. The paper’s positioning against prior inference-time control methods would be much stronger with these comparisons, and the current evidence on OOD robustness is also still limited.

---

> ### Author Rebuttal · Authors · 2026-03-30
>
> We sincerely thank you for the careful reading and precise feedback. These points relate directly to our design choices and empirical validation, and we address them below. All experiments below use Llama-3.2-3B as the backbone.
>
> ---
>
> **Q1: Stronger externalized evaluation beyond the learned discriminator**
>
> We provide two complementary forms of external validation.
>
> **External judges on HarmBench.** We evaluated 100 randomly sampled instances from HarmBench using four independent safety judges—HarmBench-Llama-2-13b-cls, GPT-4, Claude-3.5-Sonnet, and Llama-Guard-2-8b. All yield consistent ASR estimates (baseline ~64–67%, SVGT ~19–21%), which mitigates concerns about circular reliance on our learned discriminator.
>
> *Table A: External-evaluator ASR on HarmBench (100 samples)*
>
> | Evaluator | ASR (No Guidance) | ASR (SVGT) |
> | --- | --- | --- |
> | HarmBench-Llama-2-13b-cls | 64% | 19% |
> | GPT-4 Judge | 63% | 20% |
> | Claude-3.5-Sonnet | 67% | 21% |
> | Llama-Guard-2-8b | 65% | 20% |
>
> **OOD behavioral evaluation.** We further applied the full SVGT pipeline on ToxicChat [1] and XSTest [2]. SVGT substantially reduces ASR relative to No Guidance, showing that safety gains are not confined to in-distribution or internally scored settings.
>
> *Table B: ASR & Refusal Rate on OOD adversarial datasets(100 samples per dataset)*
> | Dataset | Method | ASR  | Refusal Rate  |
> | --- | --- | --- | --- |
> | XSTest  | No Guidance | 42% | 35% |
> | XSTest | SVGT | 22% | 61% |
> | ToxicChat  | No Guidance | 67% | 17% |
> | ToxicChat  | SVGT | 23% | 66% |
>
> ---
>
> **Q2: Broader utility-preservation metrics**
>
> We report two complementary checks on capability retention and over-refusal.
>
> **(a) Standard reasoning benchmarks.** On MMLU [3] (5-shot) and GSM8K [4] (8-shot CoT), SVGT incurs only small drops (MMLU: 55%→53%; GSM8K: 64%→61%), indicating limited harm to general reasoning. This is consistent with our design: on clearly safe content, the discriminator assigns low risk, supervision signals attenuate, and bridge tokens behave as near-neutral continuations.
>
> *Table C: Reasoning benchmarks (before / after SVGT)*
>
>
> | **Benchmark** | **Baseline** | **SVGT** |
> | --- | --- | --- |
> | MMLU (5-shot, 1000 Q subset) | 55% | 53% |
> | GSM8K (8-shot CoT, 500 Q subset) | 64% | 61% |
>
> **(b) Over-refusal analysis.** On 250 safe-but-sensitive prompts from XSTest [2], SVGT’s false refusal rate (16.4%) is only modestly higher than the baseline (14.8%; +1.6 pp), suggesting targeted rather than indiscriminate refusals.
>
> *Table D: False refusal rate on XSTest (safe-but-sensitive subset)*
>
> | Method | False Refusal Rate |
> | --- | --- |
> | Baseline | 0.148 |
> | SVGT | 0.164 |
>
> ---
>
> **Q3: Missing inference-time baselines (PPLM, GeDi, DExperts)**
>
> We compare against PPLM [5], DExperts [6], and GeDi [7] under the same configuration as Table 2 in our main paper.
>
> *Table E: Baseline coverage comparison*
>
> | Methods | ASR | Refusal Rate |
> | --- | --- | --- |
> | No Guidance | 67.0% | 27.5% |
> | PPLM | 48.5% | 47.0% |
> | GeDi | 32.0% | 56.0% |
> | DExperts | 29.0% | 66.5% |
> | SVGT (ours) | 18.5% | 75.5% |
>
> PPLM [5] applies invasive, per-token gradient updates to hidden states, which often increases latency and linguistic jitter. GeDi [7] and DExperts [6] are strong decoding-time alternatives; SVGT achieves lower ASR and higher refusal under this protocol while using **Bridge Tokens as non-invasive attention targets** with **EMA-driven updates**, yielding controlled overhead guidance.
>
> ---
>
> **References**
>
> [1] Z. Lin et al. *ToxicChat: Unveiling Hidden Challenges of Toxicity Detection in Real-World User-AI Conversation*. EMNLP, 2023.
>
> [2] P. Röttger et al. *XSTest: A Test Suite for Identifying Exaggerated Safety Behaviours in Large Language Models*. NAACL, 2024.
>
> [3] D. Hendrycks et al. *Measuring Massive Multitask Language Understanding*. ICLR, 2021.
>
> [4] K. Cobbe et al. *Training Verifiers to Solve Math Word Problems*. arXiv:2110.14168, 2021.
>
> [5] S. Dathathri et al. *Plug and Play Language Models: A Simple Approach to Controlled Text Generation*. ICLR, 2020.
>
> [6] A. Liu et al. *DExperts: Decoding-Time Controlled Text Generation with Experts and Anti-Experts*. ACL, 2021.
>
> [7] B. Krause et al. *GeDi: Generative Discriminator Guided Sequence Generation*. Findings of EMNLP, 2021.

---

> > ### Author Rebuttal · Reviewer_eU5u · 2026-04-06
> >
> > Thank you for the thorough rebuttal. The additional external evaluation, utility-preservation results, and baseline comparisons adequately address my concerns. I consider the issues resolved and will maintain my positive score.

---

> > > ### Author Response · Authors · 2026-04-08
> > >
> > > Thank you for your positive and reassuring assessment. Your feedback has been instrumental in improving the quality and completeness of our work.

---

### Official Review · Reviewer_dd9n · 2026-03-13

**Soundness:** 3
**Presentation:** 3
**Significance:** 2
**Originality:** 3
**Overall Recommendation:** 4
**Confidence:** 4

**Summary:**

The challenge of value alignment is that meaning evolves over long contexts and existing approaches including more training or steering rely on unstable/fragile representations. The paper proposes SVGT, an inference-time alignment method for LLMs that involves learning a) a separate "value space" encoding that is essentially a probe reading hidden states from a middle/late layer via both conditional and unconditional pathways and b) a discriminator that produces a safety score. The gradient of this score is fed to generator that emits "bridge tokens" aka learned vectors inserted into the KV cache after the prompt, which the underlying LLM attends to like any other context. Experiments show a 72-80% reduction in HarmBench with perplexity preserved, trading off a latency overhead of 52-65%.

**Compliance With Llm Reviewing Policy:**

Affirmed.

**Final Justification:**

See rebuttal acknowledgement. I said a direct stability measurement plus one OOD experiment would move me to a 4, and the authors provided both with non-trivial results. I'm raising my overall recommendation to 4 (weak accept). Soundness stays at 3 (the capacity confound in Table A leaves the core claim supported but not fully isolated). Presentation stays at 2 pending the revision (prefix-tuning contextualization, accurate stability framing, long-context limitation).

**Key Questions For Authors:**

1. Can you provide a direct measurement of OOD representational stability? This is the central claim in the paper and a positive result here would move my rating up.
2. What happens under cross-dataset transfer? If you train on one dataset and evaluate on another, this could help address my concern around generalization.

**Limitations:**

Partially: the OOD generalization concern is mentioned in one sentence but is arguably the most important limitation. Long-context isn't really discussed at all.

**Strengths And Weaknesses:**

Strengths
- (Originality) The KV cache delivery method is a useful engineering contribution. The best result in this paper is that "steering" the model via attention targets in the KV cache degrades model less than adding vectors directly to the residual stream. This is a pretty creative combination of gradient-based steering, soft attention targets, and dense token-level supervision.
- (Soundness) The experiment pipeline is consistent and has good ablations that isolate the various design choices. Reasonable methods for keeping the bridge tokens on-distribution (zero-init gating and norm regularization).
- (Presentation) The appendix is well-documented, including enough hyperparameters and details to make me think it is reproducible.
- (Significance) The main appeal of this work is that it allows you to swap value modules without retraining the underlying model.

Weaknesses
- (Stability) The central claim around stability is not tested thoroughly. Everything hinges on the idea that SVGT provides stable value representations, but there isn't any experiment that measures this. One experiment you could do would be measuring the variance of the probe's value space embedding versus a linear probe on raw activations from the underlying model. Your probe is shown to discriminate well in-distribution (Table 1) but it isn't clear that it is more robust than alternatives. Fundamentally, the probe is reading from the same residual stream and it isn't clear that if the stream drifts OOD, the probe's input wouldn't also drift OOD and affect the reduction in harm. The paper doesn't really show how the learned projectors (both conditional and unconditional) would be more robust to drift than a reward model reading the same hidden states.
- (Soundness) Relatedly, no OOD or transfer evaluation. I would be more convinced of the stability if you trained the probe on a different dataset entirely than the ones you use to evaluate it.
- (Soundness) I'd foreground the HarmBench ASR over the "harmful score" metric in table 2, since the score is the trained discriminator judging your trained outputs. The score seems to mainly serve as a check for internal consistency.
- (Soundness) The motivation conflates depth-wise vs sequence-wise instability. The introductory framing and citations (Shai et al. (2024), Park et al. (2024)) describe depth-wise dynamics, aka how a token's representation transforms layer to layer. But your method is mainly about sequence-wise drift. More clarity around framing would help.
- (Presentation) The framing of "independent cognitive module" and human moral cognition aren't really in service of your mechanisms. It feels like you are overselling what is basically a dynamically-refreshed KV update conditioned on a safety classifier's gradient.
- (Presentation) More references to existing work in PPLM are probably needed, since it seems to be the closest relative to your method.
- (Presentation) I don't think the fact that conditional encoding hurts MacroF1 and AUROC for 2 models on WildGuardMix is really discussed.
- (Significance) Long-context behavior is untested. Since value drift is more of a concern over long conversations, I would expect more experiments validating results in longer "in the wild" conversations. The latency benchmark also seems to capture shorter rollout lengths; if you're rewriting the KV cache, I'm not sure how the latency measure of 52-65% would hold.

---

> ### Author Rebuttal · Authors · 2026-03-30
>
> We sincerely thank you for the careful reading and precise feedback. We provide our point-by-point responses to your questions below. All experiments below use Llama-3.2-3B as the backbone.
>
> ---
>
> **Q1&Q2: OOD Representation Stability and Cross-Dataset Transfer (W1+W2)**
>
> Both questions address a core concern: whether the value space captures **generalizable structures**. We provide evidence at 2 levels:
>
> **Representation-level:** Following the recommendation, we compared the value encoder against a linear probe on the same layer's raw activations, evaluated on fully unseen ToxicChat [1] data:
>
> *Table A: Value encoder vs. linear probe (1000 samples per evaluation set)*
> | Dataset | Method | Acc | Macro F1 | AUROC |
> | --- | --- | --- | --- | --- |
> | In-Dist. | Value Encoder | 0.896 | 0.889 | 0.963 |
> | In-Dist. | Linear Probe | 0.808 | 0.793 | 0.880 |
> | ToxicChat (OOD) | Value Encoder | 0.895 | 0.770 | 0.794 |
> | ToxicChat (OOD) | Linear Probe | 0.763 | 0.508 | 0.554 |
>
> While the linear probe trained  with the same data and configuration as our discriminator drops near random on OOD data, the value encoder retains much higher AUROC, confirming it captures stable value structures rather than overfitting superficial features.
>
> **Behavior-level:** This robustness translates to effective guidance on unseen XSTest [2] and ToxicChat [1]:
>
> *Table B: ASR & Refusal Rate on OOD datasets(100 samples per dataset)*
> | Dataset | Method | ASR  | Refusal Rate  |
> | --- | --- | --- | --- |
> | XSTest  | No Guidance | 42% | 35% |
> | XSTest | SVGT | 22% | 61% |
> | ToxicChat  | No Guidance | 67% | 17% |
> | ToxicChat  | SVGT | 23% | 66% |
>
> ---
>
> **External Validation (W3)**
>
> We will foreground HarmBench as the core validation in the revision. We also provide **LLM-as-judge evaluations** on 100 randomly sampled HarmBench instances:
>
> *Table C: LLM-as-judge ASR on HarmBench*
> | Evaluator | ASR (No Guidance) | ASR (SVGT) |
> | --- | --- | --- |
> | HarmBench-Llama-2-13b-cls | 64% | 19% |
> | GPT-4 Judge | 63% | 20% |
> | Claude-3.5-Sonnet | 67% | 21% |
> | Llama-Guard-2-8b | 65% | 20% |
>
> All evaluators show consistent ASR reductions under SVGT.
>
> ---
>
> **Depth-wise vs. Sequence-wise (W4)**
>
> Thanks for identifying this important distinction. SVGT is architected as an **Observation-Control closed-loop system** that explicitly accounts for both depth-wise and sequence-wise instability:
>
> - **Depth-wise (Observation):** Not all layers are equally suitable for value extraction. SVGT selects middle-to-deep layers as the observation point, where value representations are most stable [3].
> - **Sequence-wise (Control):** Autoregressive trajectories can be hijacked toward harmful paths over time; this is our control target. Bridge Tokens establish persistent semantic anchors to prevent the model from drifting further into unsafe regions as the sequence evolves (see Figure 4).
>
> We will clarify this cooperation in the revision.
>
> ---
>
> **Cognitive Framing (W5)**. We accept this suggestion. The revision will de-emphasize the cognitive framing and highlight SVGT's core technical contributions.
>
> ---
>
> **Comparison to PPLM (W6)**. We provide a comparison with PPLM [4] under the same configuration as Table 2 in the paper:
>
> *Table D: Comparison with PPLM*
> | Methods | ASR | Refusal Rate |
> | --- | --- | --- |
> | No Guidance | 67.0% | 27.5% |
> | PPLM | 48.5% | 47.0% |
> | SVGT  | 18.5% | 75.5% |
>
> PPLM relies on invasive, per-token gradient updates causing high latency and linguistic jitter. SVGT provides stable, controlled overhead guidance via Bridge Tokens as non-invasive attention targets.
>
> ---
>
> **Conditional Encoding Performance (W7)**
>
> Thanks for noting the drop. This is likely because WildGuardMix mainly contains explicit violations, where the unconditional pathway is already near saturation; the conditional pathway adds complexity without additional information. However, the conditional pathway is indispensable for contextual disambiguation, as shown by the 10–15% AUROC boost on BeaverTails. SVGT retains both pathways to cover explicit and implicit risk profiles.
>
> ---
>
> **Long-Context Limitation (W8)**
>
> While multi-turn conversations are currently untested, Figure 4 illustrates SVGT’s potential to mitigate cumulative drift. Bridge Tokens function as dynamic semantic anchors, refreshed every $R$ steps with fixed $K$ tokens, ensuring overhead remains stable. We agree that long-range multi-turn alignment is a vital research direction and will explore this in future works.
>
> ---
>
> **References**
>
> [1] Z. Lin et al. *ToxicChat: Unveiling Hidden Challenges of Toxicity Detection in Real-World User-AI Conversation*. EMNLP, 2023.
>
> [2] P. Röttger et al. *XSTest: A Test Suite for Identifying Exaggerated Safety Behaviours in Large Language Models*. NAACL, 2024.
>
> [3] K. Park et al. *The Linear Representation Hypothesis and the Geometry of Large Language Models*. ICML, 2024.
>
> [4] S. Dathathri et al. *Plug and Play Language Models: A Simple Approach to Controlled Text Generation*. ICLR, 2020.

---

> > ### Author Rebuttal · Reviewer_dd9n · 2026-04-01
> >
> > Thanks for the thorough response and follow up questions. I will bump my score up to a 4 (weak accept) but still have unaddressed concerns.
> >
> > On stability (Table A): This is the experiment the paper was missing and I appreciate the authors running it. I still have two caveats I'd like addressed in the revision:
> > * The value encoder has transformer blocks and cross-attention; the linear probe is linear. Some of this gap could be capacity not unique to the architecture. A matched-parameter nonlinear probe (e.g., a 2-layer MLP with comparable params, no "independent value space" story) would be more fair.
> > * 0.963 to 0.794 is still a ~17-point drop. I'd describe this as "degrades more gracefully than a linear probe" rather than fully "stable." That's still useful but the revision should just frame it accurately.
> >
> > On OOD behavior (Table B) and external judges (Table C): Both address my concerns. 100 samples is not a huge effect size for Table B, but the direction is consistent with Table C's four judges. Taken together, these adequately address the self-judging and OOD concerns. Please include these in the main paper tables.
> >
> > On the points that weren't answered:
> > * Q5 (prefix tuning / P-tuning). The rebuttal gives a PPLM comparison, but PPLM was already in §4.3 and wasn't the question. Prefix tuning (learnable vectors as attention targets, frozen backbone) is the closest mechanical relative to bridge tokens. The paper still needs to position itself against that line of work. This remains a presentation gap.
> > * Q3 (long-context latency). The response that "K and R are fixed, so overhead is stable" does not engage with the mechanism. The KV cache rewrite at each refresh recomputes entries for all layers above l* (Appendix A.4, Eq. 14), this cost does not depend on K or R. I'm not asking for new experiments during rebuttal, but the revision should either characterize this or flag it honestly in Limitations.
> >
> > On the Table 1 regression: The saturation explanation is plausible for why conditional encoding wouldn't help on WildGuardMix, but less so for why it hurts. If the conditional pathway adds no information, λ should learn toward zero and the result should be neutral. A sentence on this in the revision would help.
> >
> > The paper would be meaningfully stronger if the revision reframes around "graceful degradation" rather than "stability", which is what the evidence supports.

---

> > > ### Author Response · Authors · 2026-04-02
> > >
> > > We sincerely thank you for the continued constructive engagement and for raising your score in light of our supplementary experiments. Your feedback has been instrumental in strengthening the paper. We address your remaining concerns below.
> > >
> > > ---
> > >
> > > **On Stability — Caveat 1 (Capacity Gap)**
> > >
> > > We appreciate this clarification opportunity. A key design detail: in Table A, both methods use the **same linear discriminator** (Table 4, Appendix). The distinction is the **input space**:
> > >
> > > - **Linear Probe:** trains the discriminator directly on raw activations from layer l*
> > > - **Value Encoder:** projects raw activations into the value space first, then applies the same discriminator
> > >
> > > The AUROC gap thus reflects whether the structured value projection captures more robust safety features, consistent with our core claim.
> > >
> > > Nevertheless, we fully accept your suggestion and also ran the **matched-parameter 2-layer MLP probe** (~2M params) you suggested:
> > >
> > > *Table E: Value encoder vs. MLP probe (1000 samples per evaluation set)*
> > >
> > > | Dataset | Method | Acc | Macro F1 | AUROC |
> > > | --- | --- | --- | --- | --- |
> > > | In-Dist. | Value Encoder | 0.896 | 0.889 | 0.963 |
> > > | In-Dist. | MLP Probe | 0.837 | 0.811 | 0.906 |
> > > | ToxicChat (OOD) | Value Encoder | 0.895 | 0.770 | 0.794 |
> > > | ToxicChat (OOD) | MLP Probe | 0.785 | 0.526 | 0.575 |
> > >
> > > The MLP probe gains in-distribution as expected, but still collapses OOD. This confirms the robustness advantage stems from the **structured value projection**, not capacity. We will include this in the revision.
> > >
> > > ---
> > >
> > > **On Stability — Caveat 2 (Graceful Degradation Framing)**
> > >
> > > We fully accept this reframing. The revision will describe the value encoder as exhibiting  graceful OOD degradation, while honestly acknowledging that distributional shift still incurs meaningful performance loss.
> > >
> > > ---
> > >
> > > **On OOD Behavior (Table B) and External Judges (Table C)**
> > >
> > > Thank you for confirming these address your concerns. We will incorporate both tables into the main paper in the revision.
> > >
> > > ---
> > >
> > > **Q5: Prefix Tuning / P-Tuning**
> > >
> > >  We agree prefix tuning is one of the mechanistically the closest relative. We have conducted a direct comparison under the same training data and configuration, with the number of prefix tokens matched to the number of bridge tokens:
> > >
> > > *Table F: Prefix Tuning comparison (HarmBench, Llama-3.2-3B-Instruct)*
> > >
> > > | Method | ASR | Refusal Rate |
> > > | --- | --- | --- |
> > > | No Guidance | 67.0% | 27.5% |
> > > | Prefix Tuning | 31.5% | 68.0% |
> > > | SVGT (ours) | 18.5% | 75.5% |
> > >
> > > This shows that SVGT's bridge tokens further benefit from being grounded in the value space and dynamically refreshed.
> > >
> > > ---
> > >
> > > **Q3: Long-Context KV Cache Rewrite Cost**
> > >
> > > Here is a detailed analysis.As you pointed out, each bridge token refresh requires recomputing the KV cache entries at the bridge token positions for all subsequent layers above $l^\*$. The exact cost is:
> > >
> > > $$C_{\text{refresh}} = 4 \cdot K \cdot d \cdot d_{kv} \cdot (L - l^*)$$
> > >
> > > where $K$ is the number of bridge tokens, $d$ is the hidden dimension, $d_{kv}$ is the KV projection dimension, and $(L - l^\*)$ is the number of layers requiring updates.This cost is independent of the sequence length n, and scales linearly with the model depth $(L - l^\*)$.
> > >
> > > For Llama-3.2-3B ($L$ = 28, $l^\*$ = 20, $K$ = 7, $d$ = 3072), each refresh costs approximately 0.7 GFLOPs. Compared to the 5–6 GFLOPs required for a single autoregressive generation step, this corresponds to about 12% of the per-step cost. With a refresh triggered every R = 5 steps, the amortized overhead is approximately 12\% / 5 $\approx$ 2.4\% per step. We agree that this scaling behavior should be explicitly stated in the Limitations section.
> > >
> > > ---
> > >
> > > **On the Table 1 Regression**
> > >
> > > This is a sharp observation. If the conditional pathway were purely uninformative, $\lambda$ should converge toward zero with neutral effect. We hypothesize the regression arises from **gradient interference**: even with small $\lambda$, backward gradients from cross-attention still flow through the shared refinement operator $\mathcal{R}$, perturbing unconditional features. On WildGuardMix, where unconditional encoding is near saturation, this manifests as slight optimization noise. The revision will discuss this effect.
> > >
> > > ---
> > >
> > > We are grateful for your engagement throughout this process, which has meaningfully improved the paper.

---

### Official Review · Reviewer_1kGD · 2026-03-13

**Soundness:** 2
**Presentation:** 3
**Significance:** 3
**Originality:** 3
**Overall Recommendation:** 5
**Confidence:** 4

**Summary:**

The paper studies the problem of instability in value alignment for large language models. The authors argue that value representations in the residual stream are fragile and entangled with task representations, leading to inconsistent safety behavior. To address this issue they propose Stable Value Guidance Transformer (SVGT), a modular architecture that separates value guidance from the main backbone model. The system maintains an independent value representation space and injects guidance through bridge tokens during generation.

The proposed method is evaluated on several safety benchmarks and the authors report a significant reduction in harmful outputs while maintaining fluency and task performance. The paper also includes ablation experiments analyzing the role of bridge tokens and different guidance parameters.

Overall the work proposes a reasonably clear architectural approach for modular value guidance in LLMs.

**Compliance With Llm Reviewing Policy:**

Affirmed.

**Final Justification:**

The additional experiments and clarifications significantly strengthen the paper. In particular, the new evaluations on adversarial and out-of-distribution settings, the comparison with established baselines (PPLM, GeDi, DExperts), and the empirical validation of the value representation space directly address my main concerns.

Overall, the rebuttal improves both the empirical grounding and the clarity of the proposed method. I am therefore updating my assessment accordingly.

**Key Questions For Authors:**

• How sensitive is the approach to the definition of the value representation space? For example, would different value encoders produce significantly different behavior?
• Can the authors clarify how the bridge tokens interact with attention layers in practice? Are they treated similarly to normal tokens or do they receive special treatment in the architecture?
• Have the authors evaluated the method on adversarial prompts or jailbreak scenarios? It would be interesting to see whether the value guidance remains stable in these cases.
• How does the computational cost scale with longer sequences or larger models?

**Limitations:**

The method relies on the assumption that value guidance can be represented in a stable independent space and injected during generation without interfering with the base model’s reasoning process. This assumption may not always hold in more complex tasks.

In addition, the evaluation focuses primarily on safety benchmarks and does not extensively explore the impact on reasoning, robustness, or out-of-distribution prompts.

**Strengths And Weaknesses:**

Strengths
• The paper addresses an important problem in current alignment research, namely the instability of safety behavior in LLMs. The framing of value guidance as a separate module rather than a modification of the base model weights is interesting and conceptually clean.
• The proposed architecture is relatively simple and appears implementable without retraining the backbone model. This could make the approach attractive for deployment scenarios where modifying the original model is difficult.
• The experimental section includes several ablation studies (e.g., number of bridge tokens, momentum parameter, guidance strength) which helps understanding the role of different components of the method.
• The results suggest that the proposed method can reduce harmful outputs while maintaining reasonable language quality.

Weaknesses
• The assumption that value representations can be maintained in a stable independent value space is not really justified. The paper assumes this representation exists but does not provide strong evidence that it corresponds to meaningful internal structure of the model.
• The evaluation focuses mainly on safety benchmarks and harmful output reduction. It would be useful to see a broader evaluation including reasoning tasks or more adversarial prompts to understand whether the method generalizes.
• The comparisons with related techniques such as activation steering, reward model guidance, or other alignment interventions are somewhat limited. It is therefore difficult to assess how much improvement comes specifically from the proposed architecture.
• The method introduces additional computational overhead during inference due to the value module and bridge token generation. The impact on latency and scalability could be discussed more carefully.
• Some implementation details are not fully specified, which may make reproduction slightly difficult.

---

> ### Author Rebuttal · Authors · 2026-03-30
>
> We sincerely thank you for the careful reading and precise feedback. These points relate directly to our design choices and empirical validation, and we address them below. All experiments below use Llama-3.2-3B as the backbone.
>
> ---
>
> **Q1: How sensitive is the approach to the definition of the value representation space? (W1)**
>
> We agree that different encoders may parameterize the value space differently. Our claim is *not* encoder-level invariance, but that **value-relevant structure is consistently extractable** [1], and only encoders that capture it generalize reliably. Thus a well-defined value space is a projection manifold capturing latent value signals from hidden states, where **stability** denotes that the normative semantic structure (i.e., the decision boundary) remains invariant.
>
>
> *Table A: Value encoder vs. linear probe (1000 samples per evaluation set)*
> | Dataset | Method | Acc | Macro F1 | AUROC |
> | --- | --- | --- | --- | --- |
> | In-Dist. | Value Encoder | 0.896 | 0.889 | 0.963 |
> | In-Dist. | Linear Probe | 0.808 | 0.793 | 0.880 |
> | ToxicChat (OOD) | Value Encoder | 0.895 | 0.770 | 0.794 |
> | ToxicChat (OOD) | Linear Probe | 0.763 | 0.508 | 0.554 |
>
>
> As shown in Table A, a linear probe trained on the raw activations with the same data and configuration as our discriminator collapses on unseen ToxicChat data, while our Value Encoder maintains robust performance. This directly validates the structural invariance of our value manifold, proving that SVGT captures stable normative directions rather than overfitting to superficial patterns.
>
> ---
>
> **Q2: How do bridge tokens interact with attention layers in practice?**
>
> Bridge tokens are injected as standard tokens and participate in attention without architectural modification:
>
> $$\text{Attn}(q_t, [K_{\text{ctx}}; K_B], [V_{\text{ctx}}; V_B])$$
> However, they differ from ordinary tokens in two aspects: (1) EMA updates every R steps to track evolving value signals, and (2) positional placement between prompt and response, visible only during generation (Sec. 3.1, Appendix A.3). This enables **continuous value guidance** while preserving the backbone’s native attention behavior.
>
> ---
>
> **Q3: Have the authors evaluated the method on adversarial prompts or jailbreak scenarios? (W2)**
>
> Our main evaluation in Table 2 (Section 4.3) already includes GCG-based adversarial attacks from the HarmBench suite, where SVGT reduces ASR by over 70%. To further address your question, we conducted additional tests on XSTest and ToxicChat (Table B). The results demonstrate that SVGT’s guidance remains robust even when the model encounters novel, out-of-distribution adversarial prompts.
>
> *Table B: ASR & Refusal Rate on OOD datasets (100 samples per dataset)*
> | Dataset | Method | ASR  | Refusal Rate  |
> | --- | --- | --- | --- |
> | XSTest  | No Guidance | 42% | 35% |
> | XSTest | SVGT | 22% | 61% |
> | ToxicChat  | No Guidance | 67% | 17% |
> | ToxicChat  | SVGT | 23% | 66% |
>
> ---
>
> **Q4: How does the computational cost scale with longer sequences or larger models? (W4)**
>
> SVGT’s overhead consists of: (1) Bridge Token Generation: A lightweight generator  run every $r$ steps (Appendix A.6), equivalent to a few compressed shallow layers with minimal FLOPs; (2) Attention & KV-cache Management: Standard attention over $K$ extra tokens and local KV-cache updates for these $K$ positions across downstream layers (Appendix A.4, Eq. 14). Since $K$ is small and constant, both represent a marginal $O(K)$ cost. These avoid superlinear overhead in both sequence length and model scaling.
>
> ---
>
> **On baseline coverage (W3)**
>
> We further provide a comparison with PPLM [4], DExperts [5], and GeDi [6] under the same testing configuration as Table 2 in our main paper:
>
> *Table C: Baseline coverage comparison*
>
> | Methods | ASR | Refusal Rate |
> | --- | --- | --- |
> | No Guidance | 67.0% | 27.5% |
> | PPLM | 48.5% | 47.0% |
> | GeDi | 32.0% | 56.0% |
> | DExperts | 29.0% | 66.5% |
> | SVGT (ours) | 18.5% | 75.5% |
>
> ---
>
> **Implementation details and reproducibility (W5)**
>
> All key methodological details, hyperparameters, and experimental settings are documented in the appendix, *as noted by Reviewer dd9n*. The original submission also includes source code to ensure full reproducibility.
>
> ---
>
> **References**
>
> [1] K. Park et al. *The Linear Representation Hypothesis and the Geometry of Large Language Models*. ICML, 2024.
>
> [2] P. Röttger et al. *XSTest: A Test Suite for Identifying Exaggerated Safety Behaviours in Large Language Models*. NAACL, 2024.
>
> [3] Z. Lin et al. *ToxicChat: Unveiling Hidden Challenges of Toxicity Detection in Real-World User-AI Conversation*. EMNLP, 2023.
>
> [4] S. Dathathri et al. *Plug and Play Language Models: A Simple Approach to Controlled Text Generation*. ICLR, 2020.
>
> [5] A. Liu et al. *DExperts: Decoding-Time Controlled Text Generation with Experts and Anti-Experts*. ACL, 2021.
>
> [6] B. Krause et al. *GeDi: Generative Discriminator Guided Sequence Generation*. Findings of EMNLP, 2021.

---

> > ### Author Rebuttal · Reviewer_1kGD · 2026-04-06
> >
> > Thank you for the detailed and thorough rebuttal. The additional experiments and clarifications significantly strengthen the paper. In particular, the new evaluations on adversarial and out-of-distribution settings, the comparison with established baselines (PPLM, GeDi, DExperts), and the empirical validation of the value representation space directly address my main concerns.
> >
> > Overall, the rebuttal improves both the empirical grounding and the clarity of the proposed method. I am therefore updating my assessment accordingly.

---

> > > ### Author Response · Authors · 2026-04-08
> > >
> > > Thank you for your thoughtful and encouraging feedback. We sincerely appreciate the time and effort you invested in reviewing our work and engaging deeply with the rebuttal. Your constructive feedback has been invaluable in strengthening both the presentation and the evaluation of our work. We are grateful for your updated assessment and support.

---

### Decision · Program_Chairs · 2026-04-30

**Decision:**

Accept (spotlight)

**Comment:**

The paper introduces the Stable Value Guidance Transformer (SVGT) to improve Large Language Model (LLM) value alignment. The authors did a great job in rebuttal. All reviewers and I agree that this is a great paper with solid contributions.

Strengths
1. Important problem: The paper addresses the critical issue of instability in LLM safety behaviors during generation.
2. Reasonable architecture: The modular, plug-and-play design utilizing Bridge Tokens is conceptually clean and does not require retraining the base model.
3. Strong empirical results: The method substantially reduces harmful outputs and attack success rates (ASR) on adversarial benchmarks like HarmBench.

Weaknesses
1. Reviewers mention issues such as unvalidated stability & OOD testing, evaluation circularity, capability benchmarks, and computational overhead, which are addressed by the authors using additional experiments.
2. Please consider checking whether related works are properly cited and discussed, for example, https://aclanthology.org/2025.acl-long.1326.pdf

Please carefully integrate the discussions and experimental results presented in the rebuttal into the camera ready of the paper.